# HDAC6 and ERK/ADAM17 Regulate VEGF-Induced NOTCH Signaling in Lung Endothelial Cells

**DOI:** 10.3390/cells12182231

**Published:** 2023-09-08

**Authors:** Sheng Xia, Heather L. Menden, Sherry M. Mabry, Venkatesh Sampath

**Affiliations:** Division of Neonatology, Department of Pediatrics, Children’s Mercy, Kansas City, MO 64108, USA; sxia@cmh.edu (S.X.); hlmenden@cmh.edu (H.L.M.); smabry@cmh.edu (S.M.M.)

**Keywords:** HDAC6, Notch signaling, SKIP, endothelial cells, ERK

## Abstract

Angiogenesis plays a critical role in various physiological and pathological processes and is regulated by VEGF. Histone Deacetylase 6 (HDAC6) is a class IIB HDAC that regulates cytoplasmic signaling through deacetylation and is emerging as a target for modulating angiogenesis. We investigated the hypothesis that VEGF-induced endothelial cell (EC) NOTCH signaling is regulated by HDAC6 through acetylation of NOTCH intracellular cytoplasmic domain (NICD). In pulmonary endothelial cells (EC), VEGF-induced activation of the NICD transcriptional response was regulated by ERK1/2 and ADAM 17 and required DLL4. While HDAC6 inhibition induced the acetylation of NICD and stabilized NICD, it repressed NICD-SNW1 binding required for the NOTCH transcriptional responses. In vitro experiments showed that HDAC6 inhibition inhibited lung EC angiogenesis, and neonatal mice treated with a systemic HDAC6 inhibitor had significantly altered angiogenesis and alveolarization. These findings shed light on the role of HDAC6 in modulating VEGF-induced angiogenesis through acetylation and repression of the transcriptional regulators, NICD and SNW1.

## 1. Introduction

Histone deacetylase 6 (HDAC6) is a class IIB HDAC that regulates protein stability and activity by deacetylating specific cytosolic non-histone substrates [1,2,3]. HDAC6 has been implicated in angiogenesis [4,5], and one of the mechanisms of its action is regulating microtubule stabilization and assembly in endothelial cells [4,5]. HDAC6 has been shown to deacetylate alpha-tubulin, which reduces the stability of the microtubule network favoring endothelial cell migration. This, in turn, can promote the formation of new blood vessels and contribute to angiogenesis [5]. In addition to regulating angiogenesis via microtubule stabilization, HDAC6 also modulates angiogenesis through the deacetylation of heat shock protein 90 (HSP90) [2,6]. HDAC6-mediated deacetylation of HSP90 enhances its chaperone activity, leading to increased stabilization of its client proteins, many of which are involved in angiogenesis [6,7]. HDAC6 repression, on the other hand, results in the acetylation of HSP90, causing misfolding and subsequent degradation of hypoxia-inducible factor 1-alpha (HIF-1α) protein, a transcription factor that is stabilized under hypoxic conditions and promotes the expression of angiogenic factors such as vascular endothelial growth factor (VEGF) [6]. Whether VEGF-induced NOTCH signaling is regulated by HDAC6 and the impact of this regulation on VEGF-induced gene expression remains unknown.

VEGF-induced Delta-like 4 (DLL4) expression activates Notch signaling and specifies endothelial cell (EC) behavior, important for angiogenesis and vascularization during embryogenesis and lung development canonically [8]. Following stimulation by Notch ligands, Notch undergoes proteolytic cleavage mediated by A Disintegrin and metalloproteinase domain-containing protein 10 (ADAM10) and γ-secretase, leading to the release of the Notch intracellular domain (NICD) into the cytoplasm [9]. The stability and activity of NICD are regulated by several mechanisms, including proteolytic cleavage, phosphorylation, and acetylation [10,11,12,13,14]. One important regulator of NICD stability is the ubiquitin–proteasome system (UPS), which is responsible for the degradation of most intracellular proteins. NICD is subject to ubiquitination by E3 ubiquitin ligases such as Fbw7, which targets NICD for proteasomal degradation. Inhibition of Fbw7-mediated ubiquitination can increase NICD stability and enhance Notch signaling [10]. In addition to proteasomal degradation, NICD stability can also be regulated by post-translational modifications, such as phosphorylation and acetylation. Phosphorylation of NICD by kinases such as glycogen synthase kinase 3 beta (GSK3β) and cyclin-dependent kinase 8 (CDK8) can decrease NICD stability and repress Notch signaling [11,12,13]. SIRT1 has been shown to directly deacetylate NICD, leading to its destabilization and degradation through the ubiquitin–proteasome system [14]. This results in decreased Notch signaling, activity and downstream target gene expression. Conversely, inhibition of SIRT1 can increase NICD acetylation and stability, leading to enhanced Notch signaling [14]. SNW1 (SNW domain containing 1) is a transcriptional co-regulator that can modulate the activity of the Notch signaling pathway by interacting with NICD and influencing its transcriptional activity [15,16]. The regulation of NICD and SNW1 activity through post-transcriptional modification and the impact of SNW1 protein acetylation on its binding to NICD have not been investigated.

Previous work from our lab had identified the key role of HDAC6 in regulating endothelial inflammation by post-translational modification of innate immune genes [17]. Further, we have recently reported that DLL4, a key downstream target of VEGF that regulates angiogenesis is required for lung vascularization in the developing lung [8]. We therefore hypothesized that HDAC6 regulates VEGF-induced NOTCH signaling through post-translational regulation of NICD and its transcriptional binding partner SNW1. In this study, we investigated the proximal mechanisms by which VEGF induced NICD activation and the regulation of NICD and SNW1 binding kinetics and transcriptional response by HDAC6 in endothelial cells (EC).

## 2. Methods and Materials

**Study Approvals:** The laboratory experiments were subjected to review and approval by the University of Missouri-Kansas City IBC, protocol number 18–28. The animal experiments were subjected to review and approval by the University of Missouri-Kansas City IACUC, protocol number 41066.

**Mouse Model:** The mice used in the experimental procedures were cared for in accordance with the guidelines set forth by the University of Missouri-Kansas City Lab Animal Resource Center and the National Institutes of Health’s regulations for the care and use of laboratory animals. All protocols were approved in advance by the University of Missouri-Kansas City Institutional Animal Care and Use Committee. The C57BL6 mouse strain was obtained from Charles River (Stillwell, KS, USA). HDAC6 inhibitor (Tubastatin-A, 1 mg/kg, SCBT, Santa Cruz, CA, USA) or buffered saline was injected intraperitoneal (ip) into C57BL6 pups on P4, P7, and P10 to inhibit HDAC6 activity. On P14, mice were euthanized with 100 mg/kg ip injection of pentobarbital followed by exsanguination via severing the brachial arteries. After cessation of the heartbeat, an angiocath was used to inflate the lungs with buffered formalin [8]. Lungs were formalin-fixed overnight, processed into slides, stained with H&E, and assessed for radial alveolar count (RAC) and mean linear intercepts (MLI) on an Olympus BX60 microscope, as previously described [8]. For all experiments, a minimum of 4 mice/group were used.

**Cell culture and reagents:** Embryonic human pulmonary microvascular endothelial cells (HPMEC) were purchased from ScienCell (Carlsbad, CA, USA) and were immortalized (HPMEC-Im) with SV40 large T antigen transformation as described before by our lab [8]. HPMEC-Im demonstrate oxidized LDL uptake, expresses classical endothelial phenotype markers, PECAM1 and ETS-related Gene (ERG), and shows angiogenic responses comparable to primary HPMEC [8]. HPMEC-Im were cultured in endothelial cell medium (ScienCell) in a humidified incubator containing 5% CO_2_ at 37 °C [8]. The following reagents were used: ERK inhibitor (U0126, 50 μM, preincubation with the cells for 2 h) was purchased from SCBT; Dibenzazepine (DBZ, YO-01027, gamma-secretase inhibitor, 10 nM, preincubation with the cells for 2 h) was purchased from Selleck Chemicals (Houston, TX, USA); KP457 (ADAM17 inhibitor, 5 µM, preincubation with the cells for 1 h) was purchased from GLPBio (Montclair, CA, USA); and recombinant human VEGF-A (VEGF, 50 ng/mL) was purchased from R&D Systems (Minneapolis, MN, USA).

**3D Angiogenesis Assay:** The 3D angiogenesis assay was carried out without fibroblasts following previously described protocols [8,18]. Briefly, HPMEC-Im treated with sc shRNA or HDAC6 shRNA coated onto cytodex microcarrier beads were embedded onto a 3D gel and cultured in 24-well plates for 5 days. There was a total of 1–3 beads per well. Roots and branches per bead were quantified using 10X images collected by Keyence microscope. Roots denote primary angiogenic tubes arising from the beads, while branches denote tubes branching from the primary roots. The length of each root from the beginning to the tip was quantified in µM and averaged.

**Luciferase Assay:** The reporter plasmids were transfected on day one, and luciferase assay was employed with Dual-Glo^®^ Luciferase Assay System (Promega, Madison, WI, USA) on the third day. Notch signaling luciferase reporter pGL4-TP1-Luc and NICD overexpression plasmid pcDNA™-DEST40-m Notch1-ICD-V5 were kindly provided by Dr. Michael Potente [14]. 

**Notch Intracellular domain (NICD) transfection:** HPMEC-Im were transfected with pcDNA3 vectors carrying wild-type NICD, a kind gift from Dr. Michael Potente at Max Planck Institute for Heart and Lung Research. HPMEC-Im were transfected using Lipofectamine 3000 following the manufacturer’s protocol. Two days after transfection, the cells were lysed for mRNA and protein expression analysis.

**shRNA mediated *HDAC6* and *DLL4* gene knockdown:** Lentivirus containing *HDAC6 shRNA* was generated and used to knock down *HDAC6* expression in HPMEC-Im (HPMEC immortalized). We used a specific shRNA (CGGTAATGGAACTCAGCACAT) that targeted human HDAC6, which has been published before [19,20]. We also verified the efficiency of our HDAC6 knockdown using Western blot analysis and PCR for HDAC6 expression, and tubulin acetylation. For the non-silenced cells, scrambled (sc) shRNA that does not interfere with cellular function was used. Knockdown efficiency was confirmed by Western blotting. DLL4 gene knockdown was conducted in a similar manner [8].

**Immunohistochemistry (IHC):** IHC staining was carried out as in a previous study [17]. The lungs of the mouse pups were inflation-fixed in 4% formalin, embedded in paraffin, and sectioned. CD31 antibody (PECAM) that outlines blood vessels was used for staining. 

**Quantification of mRNA expression using quantitative reverse transcription PCR (qRT-PCR):** Total RNA was extracted and qRT-PCR was carried out as in the previous study [8]. Primer sequences were obtained from Harvard primer bank and *18S* was used as housekeeping genes. The relative gene expression was calculated using the Pfaffl method [8].

**Immunoblotting for quantifying changes in protein expression:** Immunoblotting was carried out following standard protocol [8]. VEGF was used at 60 min for TLR4 signaling events and 24 h VEGF was used for NOTCH/angiogenesis markers. Antibodies are as follows: rabbit anti-NICD (#4147, Val^1744^, Cell Signaling, Danvers, MA, USA); rabbit anti-DLL4 (ab7280, Abcam, Waltham, MA, USA); rabbit anti-pERK (#4370, Thr^202^/Tyr^204^, Cell Signaling); rabbit anti-ERK (#4695, Cell Signaling); rabbit anti-HDAC6 (ab1440, Abcam); mouse anti-Acetylated α-Tubulin (Ac-TUBA, sc-23950, SCBT); and mouse anti-β-Actin (Clone AC15, ACTB, Sigma, St. Louis, MO, USA). Densitometry was performed using ImageJ Software (NIH, Bethesda, MD, USA) and changes were normalized to β-Actin or the corresponding non-phosphorylated antibody.

**Co-immunoprecipitation studies for detecting molecular complexes and acetylation quantification:** The co-immunoprecipitation protocol was described previously [17]. Briefly, after a 60 min VEGF treatment to the cells and lysate purification, 500 µg protein of purified protein lysate was incubated with the primary antibodies conjugated with protein A/G magnetic beads (Bio-Rad, Hercules, CA, USA). Proteins were separated by SDS-PAGE and immunoblotted with specific antibodies. Antibodies used were as follows: rabbit anti-NICD (#4147, Val^1744^, Cell Signaling); rabbit anti-acetylated Lysine (AcK, #9814, Cell Signaling); rabbit anti-HDAC6 (ab1440, Abcam); and rabbit anti-SNW1 (PA5-119879 Thermo Fisher, Waltham, MA, USA). 

**Chromatin Immunoprecipitation (ChIP):** In HPMEC-Im, we followed the Pierce Magnetic ChIP Kit’s protocol for human samples and the mouse samples followed the MAGnify Chromatin Immunoprecipitation System’s protocol (Thermo Fisher). A rabbit anti-SNW1 antibody (PA5-119879, Thermo Fisher) was used to pull down SNW1 binding to DNA, and the immunoprecipitated DNA was analyzed by PCR using specific primers corresponding with HES1 and HEY2 promoter primers that cover the RBPJ binding sites. The primers for human samples were HES1-F 5′-CCTCCCATTGGCTGAAAGT -3′; HES1-R 5′-CGGATCCTGTGTGAT CCCTA-3′; HEY2-F 5′-CGCAGGGGTTAGCAAGATTG-3′; and HEY2-R 5′-TGGTACCCCAGAGC-3′.

**Statistical Analysis:** The presented data are expressed as either mean ± SD or median with interquartile range. A significance level of *p* < 0.05 was deemed statistically significant. To quantify the data obtained from cell culture experiments, we utilized data from a minimum of three independent experiments. Littermate controls were used to obtain all animal data, and a minimum of four animals were used for each experimental group. For histological quantification, two slides per mouse were utilized, and a minimum of 4 mice/group were used. RNA quantification and PCR results were based on two to three technical replicates. Prior to conducting statistical analysis, we assessed whether the data distribution was Gaussian using the D’Agostino–Pearson omnibus normality test. If the data were normally distributed, ANOVA with a post-hoc Tukey test was used for analysis. If the data did not meet Gaussian assumptions, a Mann–Whitney U test was used for analysis. In most cases, fold changes were calculated relative to expression/changes in untreated controls. GraphPad Prism 7.0 (San Diego, CA, USA) was used for all statistical analyses.

## 3. Results

**VEGF induces NICD expression through ERK1/2, DLL4, ADAM, and **γ**-secretase in HMPEC-Im:** To investigate the rapid effect of VEGF on NICD expression, we conducted experiments using HPMEC-Im cells. We treated the cells with VEGF (50 ng/mL) for 30 min to assess NICD formation through immunoblotting. Our results revealed a significant upregulation of NICD levels after 30 min of VEGF treatment (Figure 1A,B). To demonstrate that NICD-induced downstream target gene expression in our model, we transduced HPMEC-Im with NICD. Classical targets of the NICD transcriptional response such as *DLL4*, Hes family BHLH transcription factor 1 (*HES1*), Hes-related family BHLH transcription factor with YRPW motif 1/2 (*HEY1/2*), and NOTCH-regulated ankyrin repeat protein (*NRARP*) were induced with NICD overexpression in HPMEC-Im (Figure 1C). Previous work has shown that ERK1/2 promotes angiogenesis induced by VEGF signaling in EC [21]. We, therefore, examined whether ERK1/2 regulates VEGF-induced NICD activation in HPMEC-Im. In HPMEC-Im treated with VEGF, VEGF induced ERK1/2 phosphorylation without changes in expression of ERK1/2 or the NOTCH ligand, DLL4 at 30 min (Figure 1D,E). Based on our findings of ERK activation and NICD formation in HPMEC-Im after VEGF stimulation, we postulated that VEGF activates Notch signaling through ERK1/2 in HPMEC-Im. To test this hypothesis, we performed experiments where we pre-treated HPMEC-Im cells with 50 μm ERK1/2 inhibitor (U0126) for 45 min before adding VEGF [22]. The ERK inhibitor (U0126) inhibited ERK1/2 phosphorylation (Figure 1D,E). We observed that VEGF-induced NICD formation at 30 min, and VEGF-induced *DLL4* and *HES1* gene expression at 5 h was suppressed by U0126 (Figure 1F). These results suggest that ERK1/2 regulates VEGF-induced NICD activation and downstream gene expression. 

Next, we aimed to investigate the dependence of VEGF-ERK-induced NICD expression on *DLL4*. To accomplish this, we created HPMEC-Im cells with stable knockdown of DLL4 [8]. We achieved >50% suppression of DLL4 protein expression (Figure 2A,B). In cells with DLL4 knockdown, there was less induction of NICD 30 min after VEGF treatment compared to control cells (Figure 2A,B). Consistent with this, there was decreased induction of the NOTCH-target genes HEY1 and HES1 5 h after VEGF treatment in HPMEC-Im with DLL4 knockdown (Figure 2C). We next investigated whether the ADAM family of metalloproteinases regulates VEGF-induced Notch signaling. We initially did a dose curve analysis that showed VEGF-induced NICD activation was suppressed by the ADAM17 inhibitor at 5 µM (Figure 2D,E). VEGF-induced NICD-regulated gene expression was also suppressed with the ADAM17 inhibitor (Figure 2F). ADAM family members are known to initiate cleavage of full-length NOTCH, but γ-secretase is required for subsequent cleavage that generates NICD. Studies using the γ-secretase inhibitor (DBZ, 100 nM) in HPMEC-Im cells showed repression of NICD formation confirming that VEGF-induced short-term NICD formation is dependent on γ-secretase (Figure 2G,H). These findings indicate that VEGF-induced rapid NICD activation is dependent on ERK1/2, ADAM17, and γ-secretase. 

**Inhibition of HDAC6 leads to increased expression of NICD but suppresses NOTCH signaling:** To investigate how HDAC6 regulates VEGF-induced NICD expression, we established cell lines of immortalized HPMEC (HPMEC-Im) using lentivirus carrying either scrambled (sc) shRNA or HDAC6-specific targeted shRNA. The expression of HDAC6 mRNA was confirmed through qRT-PCR (Figure 3A). VEGF-induced expression of NICD-regulated genes, such as *HES1* and *HEY1*, were repressed in cells with decreased HDAC6 (Figure 3A). We treated HPMEC-Im expressing sc shRNA or HDAC6 shRNA with VEGF for 30 min and observed an increase in NICD protein in response to VEGF (Figure 3B,C). Importantly, HPMEC-Im cells with treated HDAC6 shRNA exhibited increased levels of NICD protein in both control and VEGF-treated conditions compared to sc shRNA HPMEC-Im cells (Figure 3B,C). These data indicate that HDAC6-inhibition stabilizes NICD but represses the transcriptional activity of NICD. Increased tubulin acetylation in HDAC6 knockdown HPMEC-Im cells (Figure 3D,E) confirmed that we had inhibition of HDAC6 function. To determine whether increased NICD could result via ERK1/2, we performed immunoblotting that confirmed greater ERK phosphorylation in HDAC6 knockdown cells compared to control cells, both with or without VEGF treatment for 30 min (Figure 3D,E).

We found the increase in NICD protein levels but a decrease in NICD-regulated gene expression with HDAC6-inhibition paradoxical. We therefore proceeded to confirm our results on whether HDAC6 knockdown repressed NICD transcriptional response (decreased NICD downstream target genes) using a luciferase assay in HEK293 cells. We found reduced NICD activity in HDAC6-knockdown HEK293 cells compared to control cells (Figure 3F), consistent with our gene expression data after HDAC6-inhibition in HPMEC-Im. The results indicate that inhibition of HDAC6 enhances NICD levels but suppresses Notch signaling indicating a complex regulatory role of HDAC6 in this pathway.

**Regulation of NICD and SNW1 Binding through HDAC6-Mediated Deacetylation:** To investigate the potential role of HDAC6 in the acetylation of NICD, we conducted co-immunoprecipitation (IP) experiments using NICD antibody to assess acetylation of NICD. The efficiency of lentiviral mediated HDAC6 silencing with shRNA in lysates used for co-immunoprecipitation experiments is shown in Appendix A. Upon VEGF stimulation, we observed an increase in the levels of acetylated NICD (Ac-NICD), accompanied by a decrease in the binding between NICD and HDAC6 in HPMEC-Im cells treated with HDAC6 shRNA compared to control cells (Figure 4A,B). Our results suggested that although NICD acetylation, which stabilizes NICD and increases its levels, was increased with suppression of HDAC6, VEGF-induced Notch signaling was unexpectedly suppressed. To investigate this further, we examined interactions between NICD and SNW1, a transcriptional co-regulator of NICD-mediated gene expression that has been shown to bind NICD and is essential for inducing gene expression [15,16]. We hypothesized that NICD hyperacetylation disrupts binding to SNW1, which is required for NICD transcriptional activity. To test this hypothesis, we performed co-immunoprecipitation of NICD and SNW1. VEGF stimulation enhanced NICD acetylation and facilitated the binding between NICD and SNW1 in control cells. However, the knockdown of HDAC6 led to hyperacetylation of NICD and reduced binding between NICD and SNW1 (Figure 4C,D). 

To further explore the mechanisms by which HDAC6 knockdown upregulates NICD acetylation and stability while diminishing NICD-SNW1 binding and Notch signaling, we investigated whether HDAC6 inhibition stimulates the acetylation of SNW1, and whether this acetylation affects the binding of NICD-SNW1. By employing co-IP studies with an antibody specific to SNW1, we examined the impact of VEGF on SNW1 acetylation. Our findings demonstrated that VEGF indeed stimulated the acetylation of SNW1. Additionally, we observed that HDAC6 knockdown resulted in increased SNW1 acetylation in HMPEC-Im, along with reduced binding between NICD and SNW1 (Figure 4E,F). To assess whether the diminished binding between NICD and SNW1 affects SNW1′s ability to indirectly bind to the promoters of NICD downstream genes, we utilized chromatin immunoprecipitation (ChIP) with an SNW1-specific antibody. Our results suggest that indirect binding of SNW1 to RBPJ DNA binding sites in the promoter regions of HES1 and HEY2 was significantly inhibited with HDAC6 knockdown in HPMEC-Im following VEGF treatment, while the binding affinity remained unaltered in response to VEGF in control HPMEC cells (Figure 4G).

Our findings suggested that NICD hyperacetylation increases protein stability and expression levels (Figure 3B and Figure 4A). However, hyperacetylated SNW1 exhibited reduced binding to NICD (Figure 4E,F), resulting in decreased SNW1 indirect binding to RBPJ DNA binding sites in the promoter regions of downstream genes regulated by NICD (Figure 4G). This observation could explain the reduction in Notch signaling and downstream gene expression in HDAC6 knockdown HPMEC-Im (Figure 3A,F).

**HDAC6 regulates pulmonary EC angiogenesis:** Through the knockdown of HDAC6, we observed the repression of Notch signaling, which plays a critical role in lung angiogenesis and vascular maintenance. To investigate the impact of HDAC6 knockdown on angiogenesis in pulmonary EC, we conducted 3D cell culture assays using HPMEC-Im cells. Our findings indicate that the deficiency of HDAC6 inhibits the initiation of root formation and EC migration, while branching remains unaffected in vitro (Figure 5A–C). To examine the impact of HDAC6 inhibition on lung EC angiogenesis and lung development in vivo we administered three doses of the HDAC6 inhibitor (Tubastatin-A, 1 mg/kg) via intraperitoneal injection (ip) to neonatal mice [17]. We discovered that mice treated with the HDAC6 inhibitor exhibited double layers of capillaries in the alveolar septa, whereas control mice displayed only a single layer of capillaries consistent with normal lung development on postnatal day 14 (Figure 5C). Change in capillary structure from a double layer to a single layer in the developing alveolar septa is a key developmental event in the formation of mature alveoli from immature alveoli with primary septa [8,23]. This suggests that inhibition of HDAC6 leads to abnormal capillary formation in the lung during development. Furthermore, we assessed mouse lung development after HDAC6 inhibition. Lung morphometry analysis carried out to evaluate the progression of lung development from the saccular to alveolar phase, revealed a decrease in radial alveolar counts and an increase in mean linear intercepts in HDAC6 inhibitor-treated mice compared to control mice (Figure 5D–F). These data provide evidence that HDAC6 is essential for lung angiogenesis, and its repression disrupts distal lung morphogenesis.

## 4. Discussion

In this study, we aimed to unravel the mechanisms by which HDAC6 regulates Notch signaling and angiogenesis in lung endothelial cells. Our results reveal that VEGF activates Notch signaling through the ERK-ADAM-ү-secretase mediated cleavage of NOTCH, leading to the release of NICD. Interestingly, this activation occurs rapidly and independently of changes in DLL4 protein expression. We demonstrate that VEGF-induced NICD acetylation and stabilization are amplified with HDAC6 inhibition, but this results in repression of NICD-transcriptional activity. We further show that both NICD and its transcriptional enhancer, SNW1, are regulated by acetylation, and hyperacetylation of these proteins with HDAC6 deficiency results in decreased NICD-SNW1 binding and repressed VEGF-induced angiogenesis. Our results identify a novel, complex role for HDAC6 in regulating NICD and SNW1, required for angiogenic response in lung EC and neonatal lung development.

We discovered that VEGF-induced Notch signaling is mediated through ERK1/2, ADAM17, and ɣ-secretase, without induction of DLL4 protein expression in the short term. Conventionally, VEGF upregulates DLL4 expression via KDR-ERK1/2 signaling, which binds the receptor Notch1 *in trans*, inducing its cleavage to form NICD through ADAM10/γ-secretase [9,21]. It is important to note that Notch signaling can also be activated non-canonically through ADAM 17, which is stimulated by ERK1/2 [24,25,26]. In this study, we employed U0126 and DBZ to effectively inhibit ERK1/2 and γ-secretase, respectively, using optimized concentrations established in our previous research [22]. Through these interventions, we unveiled that the rapid activation of Notch signaling by VEGF is reliant on ERK1/2 and γ-secretases. However, we observed a reduction in Notch signaling when DLL4 was knocked down using DLL4 shRNA, suggesting the requirement of DLL4 in this process. To specifically target ADAM17, we used an ADAM17 inhibitor, KP-457, as before [25]. In our experiments, we employed 5 µM KP-457 based on dose curve experiments that showed repression of ADAM17 activity. ADAM17 repression with KP-457 resulted in decreased NICD formation and downstream target gene expression. In future studies, we aim to further confirm the role of ADAM17 in this process by utilizing ADAM17 shRNA to knock down its expression. These results suggest rapid NICD acetylation induced by VEGF can be mediated by ERK1/2-ADAM17 signaling and DLL4′s role appears to be permissive. As HPMEC-Im lysates were used for our assays the relative contribution of *cis* (cell-autonomous ERK1/2-ADAM17-NOTCH cleavage) vs. *trans* (cell-non-autonomous DLL4-NOTCH transactivation) mechanisms to VEGF-induced NICD formation and NOTCH signaling is not clarified. We speculate that both mechanisms contribute to VEGF-induced angiogenic responses, and relative contribution might depend on the acute vs. late phase responses.

Previous research has highlighted the significance of acetylation and deacetylation in regulating the stability and activity of NICD [14,27]. SIRT1, a cytoplasmic protein deacetylase, has been shown to inhibit sprouting angiogenesis by enhancing Notch signaling in zebrafish and mouse models [14]. In this study, we focused on exploring the role of another deacetylase, HDAC6, in the context of pulmonary EC angiogenesis. We observed that repression of HDAC6 resulted in the hyperacetylation and stabilization of NICD in HPMEC-Im cells. Intriguingly, the hyperacetylated NICD did not enhance Notch signaling or upregulate downstream gene expression in pulmonary EC. SNW1 has emerged as a critical regulator of Notch signaling, exerting its influence as a key coactivator within the Notch transcriptional complex. Its role encompasses the activation and modulation of Notch target genes [15]. By directly interacting with the Notch intracellular domain (NICD), SNW1 plays a pivotal part in facilitating the recruitment of coactivators and chromatin remodeling complexes to the target gene promoters [16]. Through this interaction, SNW1 enhances the transcriptional activity of NICD, leading to the expression of downstream target genes involved in diverse cellular processes [15,16]. Notably, the interplay of enhancers and repressors, SNW1, NICD, and RBPJ mediates promoter repression or activation via chromatin remodeling. However, the displacement of RBPJ repression by SNW1 facilitates the activation of the NICD transcriptional response [16,28]. This led us to hypothesize that acetylation of SNW1 might modulate its interaction with NICD and subsequently influence Notch signaling in lung EC. To test this hypothesis, we investigated the acetylation status of SNW1 in response to VEGF stimulation. Our results revealed that VEGF induced hyperacetylation of SNW1, and this hyperacetylation was associated with reduced binding between SNW1 and NICD in HDAC6-knockdown HPMEC-Im cells compared to control cells. This impairment in SNW1-NICD binding led to the repression of Notch signaling in HDAC6-knockdown cells. It is worth noting that both SIRT1 and HDAC6 might regulate the acetylation of NICD in EC. However, they exhibit opposite effects on Notch signaling regulation. The specific expression and activity of SIRT1 in pulmonary EC, as well as its potential deacetylation role on NICD and SNW1, remain unexplored avenues for future investigation.

We next examined the functional consequences of HDAC6 knockdown on pulmonary EC angiogenesis. Utilizing a 3D cell culture assay, we demonstrated that HDAC6 deficiency impaired angiogenesis in HPMEC-Im cells in vitro, which is consistent with other studies [29]. Subsequently, we extended our investigation to an in vivo mouse model, using an HDAC6 inhibitor to study the importance of HDAC6 to mouse lung development. HDAC6 inhibition resulted in abnormal capillary formation and disrupted alveolarization in the neonatal lung. These findings support the crucial role of HDAC6 in regulating pulmonary EC angiogenesis both in vitro and in vivo. While our in vivo data provide proof of concept data that support a role for HDAC6 in lung development, because HDAC6 is expressed in several lung cell types including fibroblasts, endothelial cells, and epithelial cells, the lung phenotype is likely a combination of HDAC6 inhibition on several cell types [30]. Further, our studies do not distinguish between the cell-autonomous and non-cell-autonomous role of HDAC6 in regulating endothelial angiogenesis. We plan on generating lung cell-type specific knockout mouse models to investigate this in the future.

## 5. Conclusions

Our study provides insights into how VEGF-induced activation of the NICD transcriptional response is regulated by canonical ERK1/2-ADAM17 signaling in lung EC. Our results reveal a novel regulatory role of HDAC6 in Notch signaling and angiogenesis through the regulation of NICD and SNW1 acetylation and cooperative binding required for the NICD transcriptional response. This study contributes to our understanding of the molecular mechanisms underlying lung angiogenesis and paves the way for future investigations on HDAC6′s role in developmental angiogenesis and lung alveolarization.

## Figures and Tables

**Figure 1 cells-12-02231-f001:**
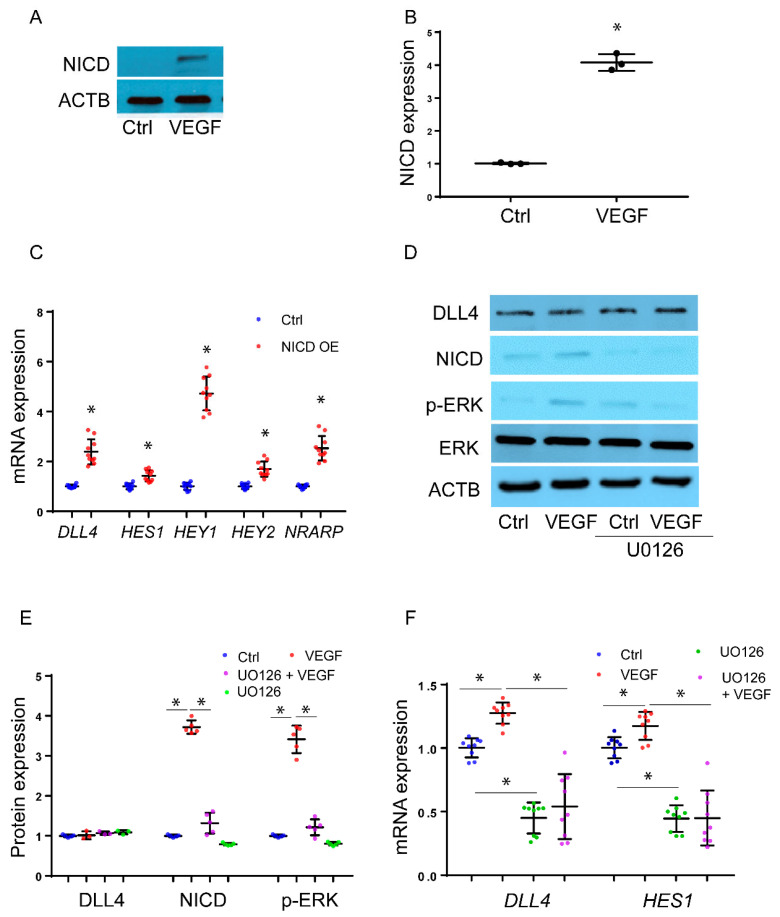
VEGF induces rapid NICD expression through ERK1/2. (**A**) NICD expression was assessed 30 min after VEGF treatment in HPMEC-Im, as shown by immunoblotting, with densitometry shown graphically; (**B**) n = 3 independent experiments, * *p* < 0.05, VEGF vs. Ctrl; (**C**) mRNA expression of *DLL4*, *HES1*, *HEY1/2*, and *NRARP* was quantified by qRT-PCR after transfection of empty or NICD-contained plasmid in HPMEC-Im, n = 10 independent experiments; * *p* < 0.05, empty plasmid vs. NICD overexpression (OE). (**D**) HPMEC-Im were treated with 50 μg/mL VEGF for 30 min with or without U0126 (ERK inhibitor) pretreatment, and DLL4, NICD, ERK, and p-ERK expression were quantified by immunoblotting, with densitometry analysis presented graphically (**E**), n = 5 independent experiments, * *p* < 0.05, VEGF vs. Ctrl. (**F**) *DLL4* and *HES1* mRNA expression was assessed by qRT-PCR 5 h after 50 μg/mL VEGF treatment. n = 9 independent experiments, * *p* < 0.05.

**Figure 2 cells-12-02231-f002:**
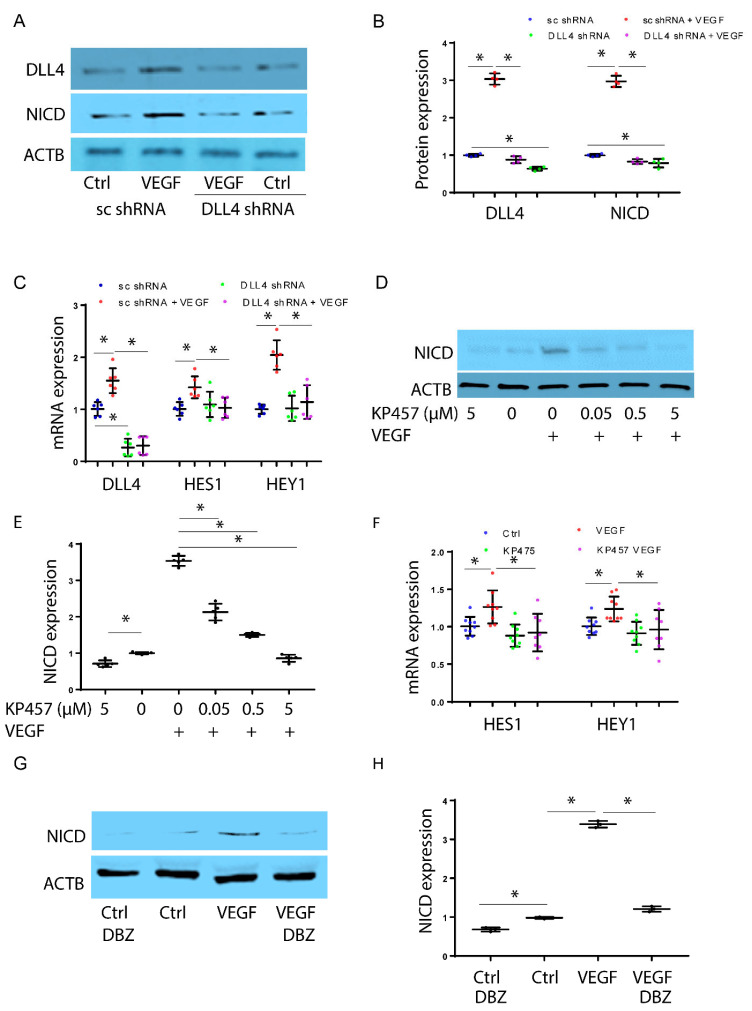
VEGF induces rapid NICD expression through DLL4, ADAM17, and ɣ-secretase. NICD expression was quantified by immunoblotting 30 min after VEGF treatment in HPMEC-Im (**A**), with densitometry analysis shown graphically (**B**), n = 4 independent experiments, * *p* < 0.05, ns means no significance. HPMEC-Im transduced with lentivirus containing sc shRNA or DLL4 shRNA were treated with VEGF, and *DLL4*, *HES1*, and *HEY1* mRNA expression was assessed by qRT-PCR (**C**), n = 6 independent experiments, * *p* < 0.05. NICD protein expression was assessed by immunoblotting after VEGF treatment with or without KP457 pretreatment (**D**). Densitometry analysis is presented graphically (**E**), n = 5 independent experiments, * *p* < 0.05. HPMEC-Im were treated with VEGF with or without KP457 pretreatment, and *HES1* and *HEY1* mRNA expression was assessed by qRT-PCR (**F**), n = 9 independent experiments, * *p* < 0.05. NICD protein expression was assessed by immunoblotting after VEGF treatment with or without DBZ pretreatment (**G**). Densitometry analysis is presented graphically (**H**), n = 3 independent experiments, * *p* < 0.05.

**Figure 3 cells-12-02231-f003:**
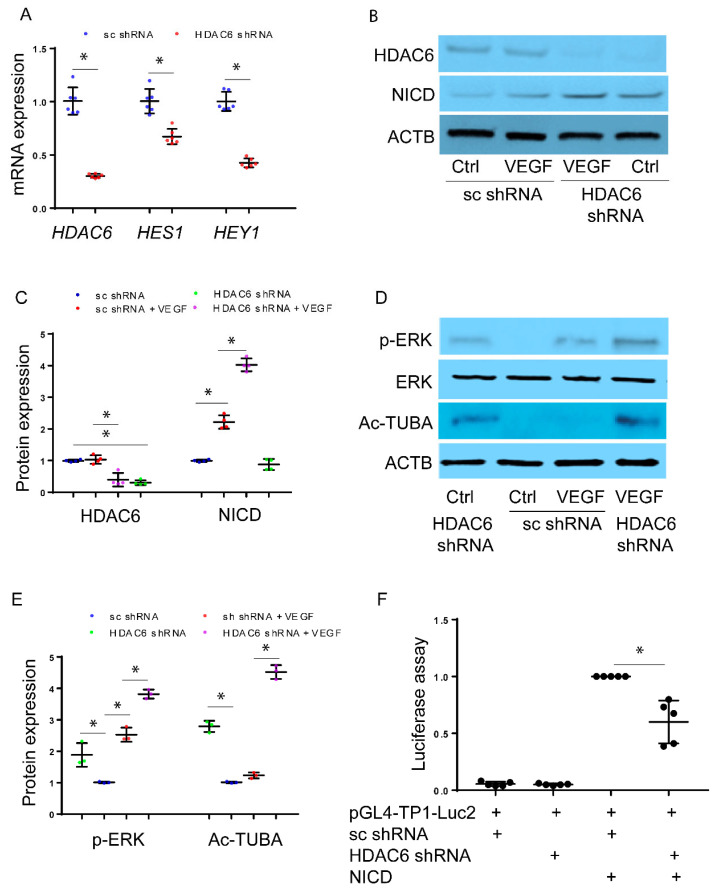
Inhibition of HDAC6 leads to increased expression of NICD but suppresses the Notch signaling pathway. HPMEC-Im transduced with lentivirus containing sc shRNA or HDAC6 shRNA. (**A**) *DLL4*, *HES1*, *HEY1,* and *JAG1* mRNA expression was assessed by qRT-PCR sc shRNA vs. HDAC6 shRNA, n = 6 independent experiments, * *p* < 0.05. (**B**) NICD protein expression was assessed by immunoblotting, with densitometry shown graphically (**C**), n = 4 independent experiments, and * *p* < 0.05. (**D**) Acetylated tubulin expression and ERK phosphorylation were assessed by immunoblotting, with densitometry shown graphically (**E**), n = 3 independent experiments, and * *p* < 0.05. (**F**) TP1-Luc2 luciferase assay was applied to assess Notch signaling activity in sc shRNA or HDAC6 shRNA-expressed HEK293 cells, n = 5 independent experiments and * *p* < 0.05.

**Figure 4 cells-12-02231-f004:**
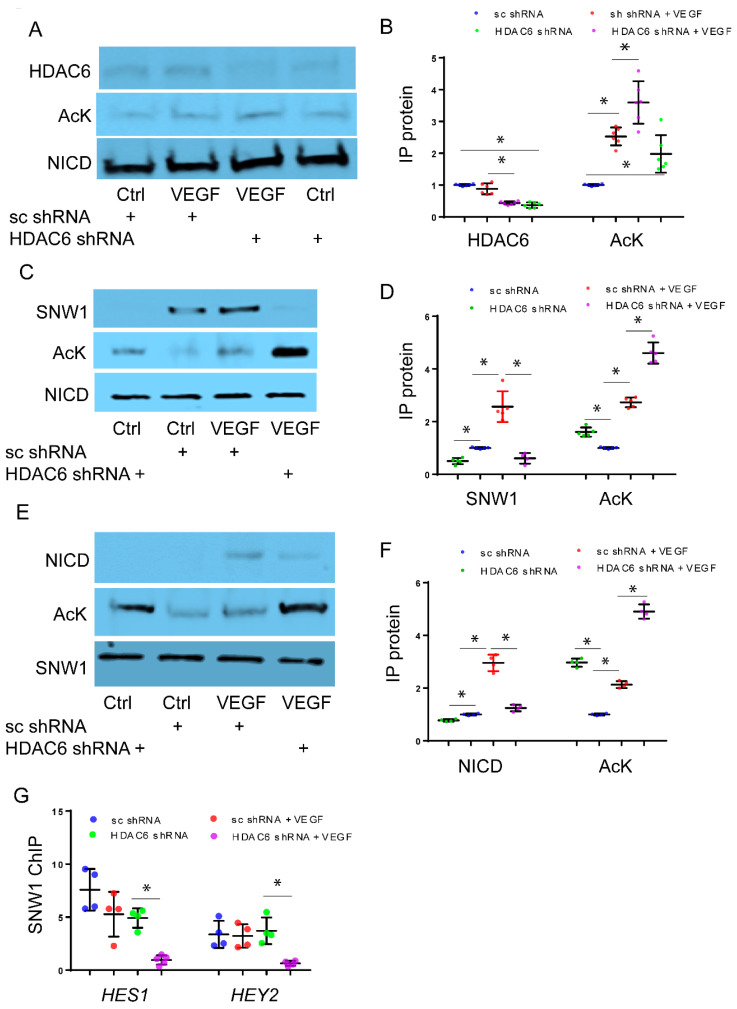
Regulation of NICD and SNW1 binding via HDAC6-mediated deacetylation. HPMEC-Im transduced with lentivirus containing sc shRNA or HDAC6 shRNA were treated with 50 ng/mL VEGF for 45 min. Immunoprecipitation (IP) with NICD antibody was performed with subsequent Western blot analysis carried out with an acetylated antibody and HDAC6 antibody on immunoprecipitated protein to quantify changes in NICD acetylation and HDAC6-NICD binding (**A**). Densitometric quantification of Western blot images in Figure 4A is shown graphically (**B**), n = 5 independent experiments; * *p* < 0.05. Similarly, NICD acetylation and NICD-SNW1 binding were assessed after IP with NICD antibody (**C**), with the results presented graphically (**D**), n ≥ 4 independent experiments, * *p* < 0.05. SNW1 acetylation and NICD-SNW1 binding were measured through immunoblotting after IP with SNW1 antibody (**E**), and densitometry analysis shown graphically (**F**), n = 5 independent experiments and * *p* < 0.05. (**G**) SNW1 binding to RBPJ binding sites on the promoters of HES1 and HEY2 was quantified using Chromatin IP (ChIP) with SNW1 antibody. n = 4 independent experiments, * *p* < 0.05.

**Figure 5 cells-12-02231-f005:**
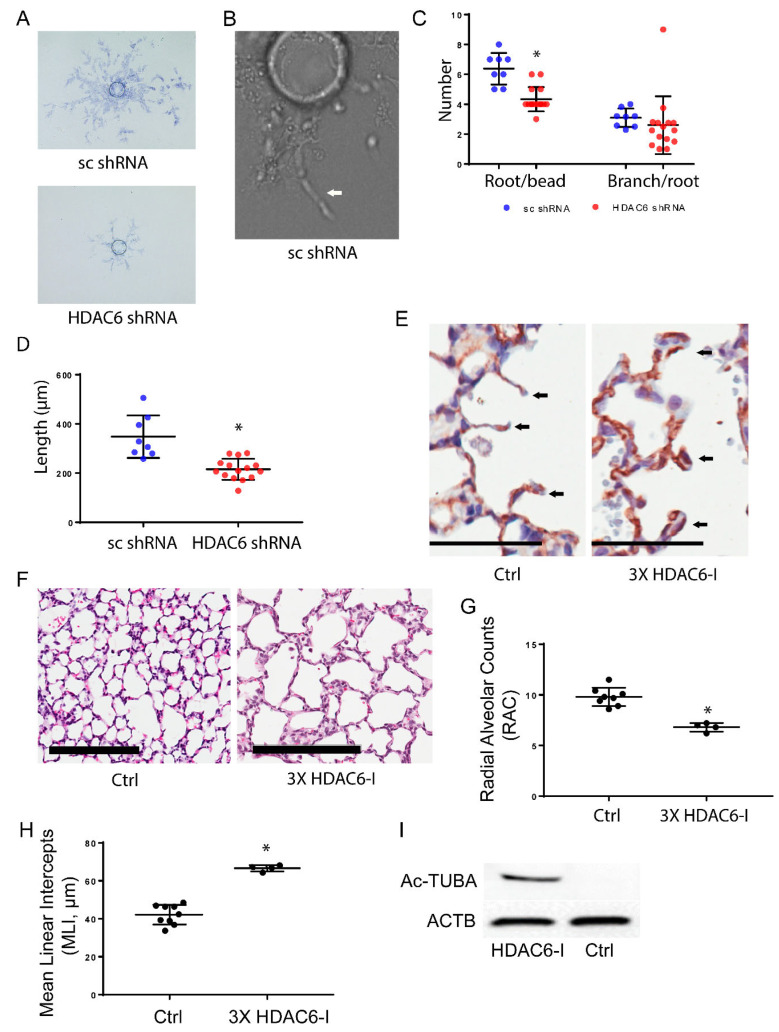
HDAC6 regulates pulmonary EC angiogenesis (**A**–**D**). - 3D angiogenesis was performed using a validated bead assay as described in the methods section in HPMEC-Im and treated with sc shRNA and HDAC6 shRNA (**A**). Magnified image of a representative sample from HPMEC-Im treated with sc shRNA showing an angiogenic tube with lumen (**B**). Quantification of roots, branches (**C**), and length of roots (**D**) is shown; 1-3 representative beads/well from 8 (sc shRNA) and 13 (HDAC6 shRNA) wells were evaluated for 3D angiogenesis. Data is derived from two independent experiments. (* *p* < 0.05). (**E**–**I**) Mouse lungs were inflation-fixed on P14 after three doses of HDAC6-I ip injection. CD31 immunohistochemical staining was performed (**E**) (scale bar: 50 µm) and H&E staining was performed (**G**) (scale bar: 200 µm) with mouse lung sections. Arrows in figure E show single vs. double layer capillaries in and around alveolar septum. Radial alveolar counts (RAC) (**G**) and mean linear intercepts (MLI) (**H**) graphically presented. The results were obtained from at least four mice per group, indicating statistical significance (* *p* < 0.05). (**I**) Western blot depicting acetylated tubulin from clarified mouse whole lung lysates.

## Data Availability

We can share data after publication with interested parties.

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
