# Peer review of "HDAC6 and ERK/ADAM17 Regulate VEGF-Induced NOTCH Signaling in Lung Endothelial Cells"

_cells, 2023, doi:10.3390/cells12182231_

Round 1

Reviewer 1 Report

This is a potentially interesting study. There are few general concerns and more specific concerns with individual figures.

General

1.     Only one targeting sequence was used. It is generally accepted for these types of studies more than one target sequence needs to be used to demonstrate that phenotype changes are not due to of target effects.

2.     Although HPMEC-lm cells are described as immortalized HPMECs in the methods, it would be helpful to the reader, if the authors would indicate that  HPMEC-lm cells are immortalized with SV40 largeT antigen in the results. The authors should provide information/ references indicating whether these largeT-immortalized HPMEC endothelial cells behave similarly to primary endothelial cells.

3.     Although the authors provide general information statistical analyses, It is important for the authors to define n in each Figure Legend or for each panel if n is different. For example, is it 3 replicas of the same experiment, 3 independent experiments,  3 independent experiments with replicas?  

4.     In the file for original images, please include all the blots used in densitometry in statistical analyses

5.     Although the authors’ in vivo data is supportive of their conclusion, it is correlative. Inhibiting HDAC6 could inhibit angiogenesis by multiple mechanisms. HDAC6 has many targets. HDAC6 inhibition could have indirect effects on angiogenesis.

Figure 1.

1.     In 1A, label lanes on blots

2.     In 1D, Replace ERK-1 with U0126, as samples were treated with ERK inhibitor and not with ERK-1.

3.     Please make this change for Fig 1E and 1F, in the labeling of conditions

4.     In 1E , please indicate the conditions as in Fig 1F. 

5.     In the legend to Figure 1 – replace ERK-1 pretreatment with the ERK inhibitor U0126

Figure 3

1.     Fig 3A would be easier to interpret if the * used to indicate significance was placed over a line indicating the conditions being compared

Figure 4

1.     For Fig 4A-B, The methods used here need to be clarified. The legend reads that NICD was immunoprecipitated and then the amount immunoprecipitated, level AcK-NICD and NICD-HDAC6 binding was determined by western blotting. Where is the data for NICD-HDAC6 binding? This data was not provided. Also, the quantitation in Fig4B indicates larger differences than observed in the representative blot in Fig4 A.

2.     Why is there such a large difference in NICD acetylation in panels A and C?

3.     Methods were not provided of chromatin IP for the data shown in 4G

Figure 5

1.     Fig 5 A-C: Little information is provided on the analysis of the bead assays used to assess in vitro angiogenesis. From the images presented in Fig. 5A, it is unclear whether these immortalized endothelial cells form sprouts and tubes, although the cells are migrating and scattering. Images need to be provided demonstrating the formation of sprouting tubes that can form branches. Also it is unclear, what the authors mean by roots.  No information is provided regarding the statistical analysis (Fig. B-C). How many experiments? How many beads?

Minor- the second and third sentences in the Introduction seem contradictory

Author Response

Reviewer 1

Comments and Suggestions for Authors

This is a potentially interesting study. There are few general concerns and more specific concerns with individual figures.

Response: Thank you for the interest in our study, and I appreciate the constructive comments the reviewer has made to improve the quality of our manuscript.

 General

  1. Only one targeting sequence was used. It is generally accepted for these types of studies more than one target sequence needs to be used to demonstrate that phenotype changes are not due to of target effects.

Response: We appreciate the reviewer’s comment. We used a specific shRNA that targets human HDAC6 (CGGTAATGGAACTCAGCACAT), which has been published before (PMID: 36810912, PMID: 20053768). We also verified efficiency of our HDAC6-knockdown using western blot analysis and PCR for HDAC6 expression, and Tubulin acetylation. While the reviewer’s comment about non-specific effects is valid, we have clearly shown inhibition of HDAC6. In the revised manuscript, we have added this to the methods section.

  1. Although HPMEC-lm cells are described as immortalized HPMECs in the methods, it would be helpful to the reader, if the authors would indicate that HPMEC-lm cells are immortalized with SV40 largeT antigen in the results. The authors should provide information/ references indicating whether these largeT-immortalized HPMEC endothelial cells behave similarly to primary endothelial cells.

Response: Appreciate the request for clarification. In the revised manuscript, we have clarified this in the methods section. We have also indicated that in our previous study we have demonstrated oxidized LDL uptake and staining for classical endothelial phenotype markers, PECAM1 and ETS-Related gene (ERG). We have referenced our previous papers describing the generation of the cell line and validation for EC characteristics.

  1. Although the authors provide general information statistical analyses, It is important for the authors to define n in each Figure Legend or for each panel if n is different. For example, is it 3 replicas of the same experiment, 3 independent experiments, 3 independent experiments with replicas?  

Response: Thank you for the guidance. We had mentioned that N represents independent experiments in the methods sections. We have also clarified that for RNA studies, a minimum of 2-3 replicates were used for each independent experiment. In the revised manuscript, we have added the N for each experiment in the legends section.

  1. In the file for original images, please include all the blots used in densitometry in statistical analyses

Response: This has been done.

  1. Although the authors’ in vivodata is supportive of their conclusion, it is correlative. Inhibiting HDAC6 could inhibit angiogenesis by multiple mechanisms. HDAC6 has many targets. HDAC6 inhibition could have indirect effects on angiogenesis.

Response: We appreciate the reviewers pointing out this important limitation. HDAC6 has potentially important roles in the endothelial cells, fibroblasts, and alveolar epithelial cells. Thus, the final phenotype shown in mouse studies is a combination of HDAC6 inhibition on several cell types. Our mouse studies provide proof of concept data supporting a role for HDAC6 in lung development. Future studies will have to clarify the cell type specific role of HDAC6 in lung development using specific knockout strategies. In the revised manuscript, we have addressed this limitation in the discussion section

 Figure 1.

  1. In 1A, label lanes on blots
  2. In 1D, Replace ERK-1 with U0126, as samples were treated with ERK inhibitor and not with ERK-1.
  3. Please make this change for Fig 1E and 1F, in the labeling of conditions
  4. In 1E , please indicate the conditions as in Fig 1F. 
  5. In the legend to Figure 1 – replace ERK-1 pretreatment with the ERK inhibitor U0126

 Response: Thank you for suggested improvements. These changes have all been made.

Figure 3

  1. Fig 3A would be easier to interpret if the * used to indicate significance was placed over a line indicating the conditions being compared

Response: Thank you for the guidance, this change has been made.

 Figure 4

  1. For Fig 4A-B, The methods used here need to be clarified. The legend reads that NICD was immunoprecipitated and then the amount immunoprecipitated, level AcK-NICD and NICD-HDAC6 binding was determined by western blotting. Where is the data for NICD-HDAC6 binding? This data was not provided. Also, the quantitation in Fig4B indicates larger differences than observed in the representative blot in Fig4 A.
  2. Why is there such a large difference in NICD acetylation in panels A and C?
  3. Methods were not provided of chromatin IP for the data shown in 4G

Response: Thank you for the comments. We have edited the legends to this figure for improved clarify. Since we immunoprecipitated NICD, the subsequent western blots would show any change in acetylation status or binding of proteins in the molecular complex. Therefore, the data shown demonstrates that HDAC6 is bound to NICD after VEGF treatments, and this decreased after HDAC6-shRNA treatment.

Since these are all independent experiments done at different times, the exact fold changes in protein are not always the same. The acetylation bands are more prominent in figure 4B when compared to figure 4A as a fresher passage of cells were used. We thank the reviewer in pointing out the variation in acetylation after HDAC6-shRNA and VEGF treatments in figure 4A. As seen in the error bar, the change in acetylation status with HDAC6-shRNA and VEGF treatments varied from 2.5 to 4.5- fold. The blot shown in figure 4A is representative of the change within the range of values.

In the revised manuscript we had added in the method for ChIP. Additionally, western blot analysis of cell lysate data of non-immunoprecipitated samples showing efficiency of HDAC6-knockdown is shown in supplementary figure 1.

Figure 5

  1. Fig 5 A-C: Little information is provided on the analysis of the bead assays used to assess in vitro angiogenesis. From the images presented in Fig. 5A, it is unclear whether these immortalized endothelial cells form sprouts and tubes, although the cells are migrating and scattering. Images need to be provided demonstrating the formation of sprouting tubes that can form branches. Also, it is unclear, what the authors mean by roots.  No information is provided regarding the statistical analysis (Fig. B-C). How many experiments? How many beads?

Response: Thank you for pointing out this oversight. In the revised manuscript, we have addressed these in methods section and in the legend to the figure. We have also provided a high-resolution image from a typical bead in HPMEC-Im treated with sc-shRNA in figure 5 of the revised manuscript. Essentially, we followed methods described by Nakatsu et al. (reference 21). Roots denote primary angiogenic tubes arising from the beads, while branches denote tubes branching from the primary roots. Length of each root from beginning to the tip was quantified in µM and averaged. Statistical information for the angiogenesis quantification is shown in the revised manuscript under the “Data analysis” section. 

Minor- the second and third sentences in the Introduction seem contradictory

Response: We appreciate the reviewer in pointing out these contradictory statements. We have edited these in the revised manuscript.

Reviewer 2 Report

Minor revision

Please the authors check through the whole text of the manuscript the spaces and the tenses of the verbs that are often used improperly.

_Introduction section, lane 2: Please the authors delete “with”.

_Methods and materials section

1) mouse models subsection, lane 7: Please the authors replace “saline” with saline buffer or saline solution.

2) Immunoblotting for quantifying changes in protein expression subsection: Please the authors standardize in the text alpha and beta symbols (for the color and for the text font).

3) Immunoprecipitation Studies for acetylation quantification subsection: Are the authors referring to the whole protein extract when they speak about the amount of 500μg? Please the authors check this point or specify.

_Results section, lane 7: Please the authors explicit the acronyms in the text when first used (HES1, HEY1/2, and NRARP).

Results, lane 34: Please the authors standardize the text font for the symbol “gamma” as in the case of alpha and beta.

Results, lane 46: Please the authors replace “sc-hRNA” with the correct form sc-shRNA.

Results, lane 58: please the author replace paradoxical (“paradoxically”) with unexpectedly.

Results, lane 76: please the authors rewrite the sentence “We hypothesized that NICD hyperacetylation might disrupt binding to SNW1 and therefore, inhibit NICD downstream expression despite increased NICD expression”, because so written it doesn't make sense, it is incomprehensible.

_Figure 5: Please the authors include the description of panel H in the capture of the figure.

_Fig 6: Please the authors rewrite the capture of the figure. Replace “Illustration showing the mechanisms by VEGF mediated NOTCH signaling is regulated by HDAC6 and ERK/ADAM17 in lung endothelial cells” with: “Illustration shows the VEGF mediated-NOTCH signaling that is regulated by HDAC6 and ERK/ADAM17 in lung endothelial cells”.

_Discussion section, lane 14: …”this process”…. to which process do the authors refer?

Discussion section, lane 16: Why the authors use Notch1 and Notch in the same sentence? Please rewrite correctly the sentence.

Discussion section, lane 33: The authors wrote “A limitation of our study is the relative contribution of cis (ERK1/2-ADAM17) vs. trans (DLL4-NOTCH transactivation) activation in VEGF-induced NOTCH signaling is not clarified”.  What the authors mean in this sentence?

Major revision

Although the work proposed by the authors for publication in Cells Journal illustrates a large amount of experiments and results aimed at demonstrating the role of Histone Deacetylase 6 (HDAC6) in the modulation of angiogenesis, it is written in a very confusing, unclear and unorganic way, leading the reader to an arduous reading of the text. Moreover, the work presents omissions that make the text sometimes incomprehensible, as in the case in which the authors discuss the results obtained in the paragraph "VEGF induces NICD expression through ERK1/2, DLL4, ADAM and É£-secretase in HMPEC-Im". Here, the authors do not specify in the text whether the cells used for their experiments (the HPMEC-Im cells) were transfected with the NICD-contained plasmid which they instead refer to in the caption of Figure 1. There is no correspondence between the text and the figure to which the text refers. Furthermore, the authors often make an improper and confusing use of the terminology of the system Notch/NICD, as in the case of the statement “VEGF-induced NICD expression”. In fact, Notch undergoes ADAM10 and É£-secretase-mediated proteolytic cleavage, upon stimulation by its ligands, leading to the release of the intracellular Notch domain (NICD) into the cytoplasm. VEGF activates Notch through Dll4, but NICD derives from the cleavage of the activated receptor, so the authors use improperly in the text  “NICD expression”. They can speak about an increase of the levels or an increase in the amount of proteolytic form of Notch receptor NICD, but not of VEGF-induced NICD expression. The authors should be more careful in writing  also the final discussion that often unclearly summarizes their results. So, I strongly suggest to the authors to rewrite the text in a more organic and clear form for the publication.

Please the authors check through the whole text of the manuscript the tenses of the verbs that are often used improperly and the construction of sentences, some of which are unclear.

Author Response

Reviewer 2

Minor revision

Please the authors check through the whole text of the manuscript the spaces and the tenses of the verbs that are often used improperly.

_Introduction section, lane 2: Please the authors delete “with”.

Response: Thank you for the suggestion. This change has been made.

_Methods and materials section

1) mouse models subsection, lane 7: Please the authors replace “saline” with saline buffer or saline solution.

Response: Thank you for suggesting this change, it has been made.

2) Immunoblotting for quantifying changes in protein expression subsection: Please the authors standardize in the text alpha and beta symbols (for the color and for the text font).

Response: Thank you for your comment. This change has been made.

3) Immunoprecipitation Studies for acetylation quantification subsection: Are the authors referring to the whole protein extract when they speak about the amount of 500μg? Please the authors check this point or specify.

Response: Thank you for requesting this clarification, which has been made in the revised manuscript.  text.

_Results section,

lane 7: Please the authors explicit the acronyms in the text when first used (HES1, HEY1/2, and NRARP).

Results, lane 34: Please the authors standardize the text font for the symbol “gamma” as in the case of alpha and beta.

Results, lane 46: Please the authors replace “sc-hRNA” with the correct form sc-shRNA.

Results, lane 58: please the author replace paradoxical (“paradoxically”) with unexpectedly.

Results, lane 76: please the authors rewrite the sentence “We hypothesized that NICD hyperacetylation might disrupt binding to SNW1 and therefore, inhibit NICD downstream expression despite increased NICD expression”, because so written it doesn't make sense, it is incomprehensible.

Response: Thank you for your comments. All these changes have been made.

_Figure 5: Please the authors include the description of panel H in the capture of the figure.

Response: Thank you for this request, it has been made.

 _Fig 6: Please the authors rewrite the capture of the figure. Replace “Illustration showing the mechanisms by VEGF mediated NOTCH signaling is regulated by HDAC6 and ERK/ADAM17 in lung endothelial cells” with: “Illustration shows the VEGF mediated-NOTCH signaling that is regulated by HDAC6 and ERK/ADAM17 in lung endothelial cells”.

Response: Thank you for this comment. In the revised manuscript, figure 6 has been changed to a graphical abstract, and the legend has been changed based on the suggestion.

 _Discussion section,

lane 14: …”this process”…. to which process do the authors refer?

Discussion section,

Response: We agree with the reviewer that this sentence is not very clear. In the revised manuscript, we have edited the first two sentences of this paragraph for improved clarity.

lane 16: Why the authors use Notch1 and Notch in the same sentence? Please rewrite correctly the sentence.

Discussion section,

Response: This has been corrected in the revised manuscript.

lane 33: The authors wrote “A limitation of our study is the relative contribution of cis (ERK1/2-ADAM17) vs. trans (DLL4-NOTCH transactivation) activation in VEGF-induced NOTCH signaling is not clarified”.  What the authors mean in this sentence?

Response: In the revised manuscript we have re-written this sentence for improved clarity. Essentially, since we used HPMEC-Im lysates for all our assays, whether VEGF-induced NOTCH signaling that results in NICD-dependent gene expression is dependent on ERK1/2-ADAM17-mediated NOTCH cleavage (cis) or DLL4-NOTCH transactivation and cleavage is not clarified. This is a limitation of the study and is discussed in the revised manuscript.

Major revision

Although the work proposed by the authors for publication in Cells Journal illustrates a large amount of experiments and results aimed at demonstrating the role of Histone Deacetylase 6 (HDAC6) in the modulation of angiogenesis, it is written in a very confusing, unclear and unorganic way, leading the reader to an arduous reading of the text. Moreover, the work presents omissions that make the text sometimes incomprehensible, as in the case in which the authors discuss the results obtained in the paragraph "VEGF induces NICD expression through ERK1/2, DLL4, ADAM and É£-secretase in HMPEC-Im". Here, the authors do not specify in the text whether the cells used for their experiments (the HPMEC-Im cells) were transfected with the NICD-contained plasmid which they instead refer to in the caption of Figure 1. There is no correspondence between the text and the figure to which the text refers. Furthermore, the authors often make an improper and confusing use of the terminology of the system Notch/NICD, as in the case of the statement “VEGF-induced NICD expression”. In fact, Notch undergoes ADAM10 and É£-secretase-mediated proteolytic cleavage, upon stimulation by its ligands, leading to the release of the intracellular Notch domain (NICD) into the cytoplasm. VEGF activates Notch through Dll4, but NICD derives from the cleavage of the activated receptor, so the authors use improperly in the text  “NICD expression”. They can speak about an increase of the levels or an increase in the amount of proteolytic form of Notch receptor NICD, but not of VEGF-induced NICD expression. The authors should be more careful in writing  also the final discussion that often unclearly summarizes their results. So, I strongly suggest to the authors to rewrite the text in a more organic and clear form for the publication.

Please the authors check through the whole text of the manuscript the tenses of the verbs that are often used improperly and the construction of sentences, some of which are unclear.

Response: We appreciate the authors suggestion on improving the writing and clarity of the manuscript in general. We have gone through the entire manuscript in detail, and made several changes to improve the clarity and use of language with particular attention to tenses and verbs. We are thankful for the guidance.

As recommended by the reviewer, we have changed NICD expression to NICD levels or formation throughout the revised manuscript.

Reviewer 3 Report

The research article “HDAC6 and ERK/ADAM17 regulates VEGF-induced NOTCH signaling in Lung Endothelial Cells.” by Xia et al., presented solid evidence of HDAC6's role in VEGF-induced angiogenesis and alveolar development in the neonatal lungs.

50 ng/ml VEGF is too high a concentration. Could be forcible effects. Comment, please?

Curious to know. Was VEGFR2 active (pVEGFR2) at 30 min and 5h after the VEGF stimulation?

The authors state that “We observed that VEGF-induced NICD expression at 30 minutes and VEGF-induced NICD-regulated downstream gene expression at 5 hours was suppressed by ERK-I (Figure 1F) Please include the list of genes that are tested in this experiment.

Please help the reader by describing the acronyms as they appear for the first time in the text. For example, ERK-I; I guess it is ERK inhibition?

It would be nice to see the consistency in the order of gel loadings in the representative western blots.

 I am surprised to see a single band of ERK/pERK. Usually, these always show up doublets at 42 and 44KD. Comment, please.

Minor

There are a couple of Western blots that are under-exposed which could be replaced

None

Author Response

Reviewer 3

Comments and Suggestions for Authors

The research article “HDAC6 and ERK/ADAM17 regulates VEGF-induced NOTCH signaling in Lung Endothelial Cells.” by Xia et al., presented solid evidence of HDAC6's role in VEGF-induced angiogenesis and alveolar development in the neonatal lungs.

50 ng/ml VEGF is too high a concentration. Could be forcible effects. Comment, please?

Response: Thank you for this comment. We had initially tried both 25ng/mL and 50ng/mL doses of VEGF, and although we could elicit similar responses with either, the 50ng/mL dose provided us with the most consistent responses.

Curious to know. Was VEGFR2 active (pVEGFR2) at 30 min and 5h after the VEGF stimulation?

Response: While we did not specifically probe VEGFR2 phosphorylation, other investigators have shown that VEGF induced ERK1/2 phosphorylation and DLL4 expression are both mediated by VEGFR2 phosphorylation. VEGFR2-phosphorylation activates NOTCH signaling that regulates NICD target gene expression.

The authors state that “We observed that VEGF-induced NICD expression at 30 minutes and VEGF-induced NICD-regulated downstream gene expression at 5 hours was suppressed by ERK-I (Figure 1F) Please include the list of genes that are tested in this experiment.

Response: We thank the reviewer for requesting this clarification. We have made the corresponding changes in the results section.

Please help the reader by describing the acronyms as they appear for the first time in the text. For example, ERK-I; I guess it is ERK inhibition?

Response: Thank you pointing out this oversight. In the revised manuscript, the acronym was described the first time in the text on page 4. We have also ensured that we have expanded acronyms at first use for others too.

It would be nice to see the consistency in the order of gel loadings in the representative western blots.

Response: Thank you for this comment. While we cannot change how the gels where loaded for all the experiments, we have provided the original blots as a PDF document.

 I am surprised to see a single band of ERK/pERK. Usually, these always show up doublets at 42 and 44KD. Comment, please.

Response: Thank you for this question. We have found the higher 44kDa when the film is overexposed, and a bit inconsistently. Therefore, we have in general not preferred to depict this.

Minor

There are a couple of Western blots that are under-exposed which could be replaced

Response: We have attached better western blots for some of these, thanks.

Round 2

Reviewer 2 Report

I'm pleased to ascertain that the authors have followed the suggestions that I recommended, aimed to improve the quality of the manuscript submitted to my attention. After revision the manuscript can now be considered fine for publication in Cells Journal.